# The Kinematics of Social Action: Visual Signals Provide Cues for What Interlocutors Do in Conversation

**DOI:** 10.3390/brainsci11080996

**Published:** 2021-07-28

**Authors:** James P. Trujillo, Judith Holler

**Affiliations:** 1Donders Institute for Brain, Cognition and Behaviour, Radboud University Nijmegen, 6525 GD Nijmegen, The Netherlands; judith.holler@mpi.nl; 2Max Planck Institute for Psycholinguistics, Wundtlaan 1, 6525 XD Nijmegen, The Netherlands

**Keywords:** kinematics, intention, social action, conversation, social interaction

## Abstract

During natural conversation, people must quickly understand the meaning of what the other speaker is saying. This concerns not just the semantic content of an utterance, but also the social action (i.e., what the utterance is doing—requesting information, offering, evaluating, checking mutual understanding, etc.) that the utterance is performing. The multimodal nature of human language raises the question of whether visual signals may contribute to the rapid processing of such social actions. However, while previous research has shown that how we move reveals the intentions underlying instrumental actions, we do not know whether the intentions underlying fine-grained social actions in conversation are also revealed in our bodily movements. Using a corpus of dyadic conversations combined with manual annotation and motion tracking, we analyzed the kinematics of the torso, head, and hands during the asking of questions. Manual annotation categorized these questions into six more fine-grained social action types (i.e., request for information, other-initiated repair, understanding check, stance or sentiment, self-directed, active participation). We demonstrate, for the first time, that the kinematics of the torso, head and hands differ between some of these different social action categories based on a 900 ms time window that captures movements starting slightly prior to or within 600 ms after utterance onset. These results provide novel insights into the extent to which our intentions shape the way that we move, and provide new avenues for understanding how this phenomenon may facilitate the fast communication of meaning in conversational interaction, social action, and conversation.

## 1. Introduction

Human conversational interaction is characterized by rapid exchanges of speaking turns between interlocutors. This fast-paced exchange of speaker turns is thought to be possible because next speakers plan their own turn in parallel to the current speaker uttering their turn. This is accomplished by next speakers predicting the meaning of the ongoing utterance, including the social action (e.g., requesting information, checking mutual understanding, etc.) that the utterance is performing. However, less is known about whether early, visual signals, such as movement of the torso, head, or hands could also play a role in this process by signaling intentions during interaction. The present study is a first attempt to glean insight into this issue.

Social actions refer to what an utterance does in conversation [1], for example questioning, responding, and stating. They are core to determining the course of an interaction, since pragmatically utterances are required to match the social actions that precede them in a sequentially organized fashion [2,3,4]. However, these broad categories, such as questioning, are quite diverse in terms of the more fine-grained interactional work that they do and may thus fall into several subcategories of social action [5]. For example, questions may be designed to function as a request for information (e.g., “what time is it?”), to initiate repair in case of misunderstandings or mishearings (e.g., “what did you say?”, “who?”), to criticize (e.g., “do you really think that was a good idea?”), etc. Of course, some questions may also function in more than one way. While there is some evidence for the body playing a role in intention signaling of broad social actions (see below), it is currently not known if we also provide visual signals that help our addressee infer these more fine-grained social actions. Investigating the role of bodily kinematics on signaling social actions during natural conversation therefore provides novel insights into the granularity at which our intentions shape the way that we move in order to communicate.

Previous research strongly suggests that instrumental movement, such as grasping a bottle in order to do something with it, can reveal and signal our intentions to others [6,7,8,9,10,11]. In particular, movement kinematics can provide early clues to one’s concrete intentions (i.e., proximal goal-oriented) and social intentions, allowing an observer to predict what the person is doing before the action is complete [12,13]. In these studies, it was found that the kinematics of the arm and hand while reaching to grasp a bottle differed in systematic ways, such as the height of the wrist at certain points in the movements, depending on whether the person was intending to grasp the bottle in order to pour from it, or in order to drink from it. In particular, kinematics occurring within the time bins at 50–80% of the total movement duration, which corresponds approximately to the first 500–800 ms of the action, were highly predictive of the concrete intention of the action (i.e., to drink, to pour [13]). Similarly, the social intention of a person also influences their kinematics. This can be seen in a study showing that the kinematics of xylophone-playing movements are exaggerated in size when the person is demonstrating a sequence for another person, or when they are coordinating with another person [7]. Kinematic exaggeration of movements that are intended for an observer have also been shown for communicatively intended iconic (silent) gestures [14,15]. Of course, in all of these cases the form of the action/gesture is the foundation for recognizing these kinematic deviations as being meaningful. These studies demonstrate that not only what we intend to do, but the intended social effect of what we are doing, both shape how we move. In the context of conversation, it could be that our movements signal the intentions underlying social actions, but this is largely uncharted territory.

The multimodal nature of human communication [16,17,18,19,20,21,22] suggests that visual signals may play a role in the signaling of social action. This is because, similar to instrumental action intentions, research suggests that social actions are processed very rapidly, during the first 1000 ms of an unfolding utterance, and in some cases considerably faster [23,24]. While it seems possible that this fast processing is facilitated by visual signals, it is currently not known if different social actions are indeed marked by differences in co-occurring movement kinematics that could be used by an observer.

To provide a first larger-scale systematic investigation into whether visual movements may be providing cues to specific social actions, we use naturalistic dyadic conversation and motion tracking data from a conversational corpus to test two main questions. We ask whether visual kinematics during questions differ depending on the specific social action that the question is serving. In the present study, we focus on six such social actions conveyed through questions which occurred frequently in the conversational data: request for information, other-initiated repair, providing an evaluation or stance, checking mutual understanding, expressing active participation, or indicating the speaker’s wondering or thinking in a self-directed manner. We here focus our analysis on a 900 ms time window (ranging from 300 ms prior to utterance onset to 600 ms post utterance onset). This choice of time-window was motivated by the fact that social action recognition based on speech alone also seems to be possible within this time frame [23,24], as well as by the fact that co-speech visual signals can start prior to utterance onset, but also some time after the utterance has begun [25,26]. Due to variation in utterance length, this means that the time window includes most of the utterance for some of them, but only the beginning for others. However, it does allow for a first insight into the potential of movement contributing to fast social action recognition within a specified timeframe for cognitive processing.

In the present analysis, we do not investigate specific movements (e.g., individual communicative gestures, types of head movements or particular postures). Instead, we focus on whether there is evidence at the level of movement more generally, in part inspired by the literature on links between instrumental actions and intentions which, too, tends to focus on generic descriptors such as movement height (or relative position) or velocity. This allows us to examine if there is any evidence in the kinematics at all, without a priori assumptions about specific movement categories. In order to do this, we calculate four basic kinematic features that characterize movements in geometric space. Movement Magnitude, which captures the three-dimensional space that an articulator has moved during our window of observation. Magnitude is taken as a simple measure of movement extent that can capture various types of movements (such as a nod, a potentially meaningful shift in posture, a manual gesture, or even moving the hands out of the lap to prepare a manual gesture). In other words, a general indicator of the extent of bodily involvement. Lateral Position, which captures movement to the left or right, such as postural shifts or sideways leans. Anterior/Posterior Position, which captures movement forward and backwards. This can relate to leaning forward, thrusting the head forward, or gesturing towards the interlocutor, but also moving oneself backward from the starting position (e.g., to “distance” oneself from what is being said). Movement Velocity, which captures the peak velocity of the articulator. This can capture visual salience of a movement, even if the movement is small. Together, these basic kinematic features can characterize even very different movements produced across different bodily articulators, capturing size (spatial salience), velocity (temporal salience), and changes in location. Due to this research charting largely novel territory, extant literature is scarce, and our analyses are primarily exploratory at this stage. However, the aim is to use these first investigations to lay the groundwork for more targeted future studies in this domain. Thus, this study aims to provide novel insights into whether there is any evidence that co-speech movement kinematics are shaped by the fine-grained intentions underlying the social actions of conversation at all.

## 2. Methods

### 2.1. Participants

The project utilized a corpus of 34 dyads (mean age: 23.10 ± 8 years; 51 females, 17 males) engaged in face-to-face interaction. Dyads consisted of pairs of acquaintances, who were all Dutch native speakers, without motoric or language problems and with normal or corrected-to-normal vision.

### 2.2. Data Collection

Dyads engaged in three recording sessions, each lasting 20-min. In the first 20 min, participants held a free, entirely unguided conversation. In the second session, participants discussed one of three themes: privacy, social media, or language in teaching. Participants were instructed to share their opinions about these themes and to discuss their agreements and disagreements per theme. Participants read examples of the themes before starting with the second task. If a dyad finished discussing one theme, they could choose another. During the third session, participants were asked to imagine their ideal holiday that is affordable on their own budget. They discussed these ideas with their partner with the aim of devising a holiday plan that they would both enjoy. These different sessions were used to encourage a variety of conversational social actions typical of interactions interlocutors often engage in (described below). Given that participants varied substantially in terms of the precise content, their perspectives, and moods, for example, even when discussing the same broad topics, we are confident that the data captured in this corpus is quite representative of everyday talk.

The conversations were recorded in a soundproof room at the Max Planck Institute for Psycholinguistics in Nijmegen, The Netherlands. Participants were seated directly facing each other.

Two video cameras (Canon XE405) were used to record frontal views of each participant, two cameras recorded each participant’s body from a 45-degree angle (Canon XF205 Camcorder), two cameras (Canon XF205 Camcorder) recorded each participant from a birds-eye view while mounted on a tripod, and finally one camera (Canon Legria HF G10) recorded the scene view, displaying both participant at the same time. All cameras recorded at 25 fps. Audio was recorded using two directional microphones (Sennheiser me-64) for each participant (see the Appendix A for an overview of the set-up). Each recording session resulted in seven video files and two audio files, which were synchronized and exported as a single audio-video file for analysis in Adobe Premiere Pro CS6 (MPEG, 25 fps), resulting in a time resolution of approximately 40 ms, the duration of a single frame. Motion tracking was captured using two Microsoft Kinect V2s for each participant, with one used to capture facial kinematics (using Brekel Pro Face 2.39) and the other for capturing body movement (using Brekel Pro Body 2.48). Only data from the body recordings are discussed here. The Kinect was used as it is able to capture movements in three-dimensional space, and has been shown to be valid for reliably quantifying hand/arm kinematics [27], torso kinematics [28], as well as head posture [29]. In order to calibrate the motion tracking devices (Kinect for Windows 2) participants held a T-pose for three seconds. A hand clap was used in order synchronize the audio and visual recordings. In order to synchronize the Kinect with the video recordings, we used a custom Python (v.3.7) script that aligns the audio recording from the video with the audio recording from the Kinect. We additionally manually checked whether the beginning and end of the synchronized audio streams indeed matched, ensuring an offset did not accumulate during the recording time. See Figure 1 for a visual overview of the recording set-up.

Informed consent was obtained before and after filming. Participants were asked to fill in a demographic’s questionnaire prior to the study, and four questionnaires at the end of the study. Information was obtained about the relationship between the conversational partners and their conversation quality, and the Empathy Quotient [30], the Fear of Negative Evaluation scale [31], and a question assessing explicit awareness of the experimental aim were also collected (the questionnaire data was not included in the present analyses since no specific hypotheses related to this study pertained to them). Information from these questionnaires is not discussed in the current study. Participation was compensated with 18 euros. The corpus study was approved by the ethics committee of the social sciences department of the Radboud University Nijmegen.

### 2.3. Data Annotation: Questions

Manual annotation first captured all questions (and responses, which are not part of the focus of the present analyses). An automatic orthographic transcription of the speech signal was made using the Bavarian Archive for Speech Signals Webservices [32]. Questions were identified and coded in ELAN (5.5; [33,34]), largely following the coding scheme of Stivers & Enfield ([35]). In addition to this scheme, more rules were applied on an inductive basis, in order to account for the complexity of the data in the corpus. Specifically, a holistic approach was adopted, taking into consideration visual bodily signals, context, phrasing, intonation, and addressee behaviour. Non-verbal sounds were excluded (e.g., laughter, sighs). This was done by two human coders, one native speaker of Dutch, and one highly proficient speaker of Dutch. Interrater reliability between the two coders was calculated with raw agreement [36,37] and a modified Cohen’s kappa using EasyDIAg [38] on 12% of the total data (4 dyads, all tasks). EasyDIAg is an open-source tool that has been used as a standard method for calculating a modified Cohen’s kappa. It is based on the amount of temporal overlap between annotations, categorization of values, and segmentation of behaviour. A standard overlap criterion of 60% was used. Reliability between the coders resulted in a raw agreement of 75% and *k* = 0.74 for questions, and a raw agreement of 73% and *k* = 0.73 for responses, indicating substantial agreement. The precise beginnings and endings of the question annotations were segmented using Praat (5.1, [39]) based on the criteria of the Eye-Tracking in Multimodal Interaction corpus (EMIC; [40,41]). This resulted in a total of 6778 questions (duration *Mdn* = 1114 ms, *range =* 99–13,145 ms, *IQR =* 1138 ms).

### 2.4. Data Annotation: Social Action Categories

From the total annotated questions, a subset of questions from each participant were additionally coded for their social action category. This resulted in a total of 2078 question utterances included in the present study. As part of the larger project the present analysis forms part of a detailed coding manual for social actions was developed, leading to a set of inductively generated categories into which all of these questions would fit. Questions may sometimes perform more than one single social action [5], but for the purpose of the present analysis, we considered the primary social action classification for each question only. For the present analysis, questions from six discrete categories were used. More fine-grained sub-categories were also defined, but these are not used in the present analyses. Information Requests (InfReq; n = 693) refer to any questions that ask for new information of a factual or specific nature (e.g., “what time is it?”, “what is this?”), or requests for elaboration or confirmation. Understanding Checks (UndCheck; n = 365) refer to requests for the interlocutor to confirm information, particularly when the speaker believes to already know the answer but wants to check (e.g., while making travel plans, “And you said you wanted to travel next week?”; referred to as CHECK questions by Jurafsky, 2003 [42]), or to confirm that the interlocutor is following the relevant information in the conversation (e.g., “you know what I mean?”). Self-Directed questions (SelfDir; n = 360) refer to questions that are not directed at the other speaker (e.g., “now where are my keys?” while looking in a bag; “… or actually, was that somewhere else?”), that may be used to fill pause and show the interlocutor that the current speaker is thinking and wants to keep the turn. Stance or Sentiment questions (StanSem; n = 246) refer to questions that express disapproval or criticism (e.g., “where do you think you’re going?”), seek confirmation or approval (e.g., “isn’t it beautiful today?”), challenges to the interlocutor to prove or justify something (e.g., “but you didn’t do that, did you?”), or corrections to something the interlocutor previously stated (e.g., “wasn’t it the other way around?”). Other-initiated-repair questions (OIR; n = 126) are used to resolve possible misunderstandings (e.g., “what do you mean?”) or mishearings (e.g., “what did you say?”) [43]. Active participation questions (ActPart; n = 161) included news receipts (e.g., A: Hey that was the same spot we took off for Honolulu. Where they put him on, at that chartered place B: Oh really?; Heritage, 1985, p. 302) that may or may not encourage elaboration, surprise expressions (e.g., “what?”, “really?” in response to surprising information), expressing disbelief, scepticism or incredulity towards what was just said (e.g., “You’re going by plane?!”, asked when the speaker cannot believe that their interlocutor is flying to a nearby city that can be easily accessed by train or car; Crespo Sendra et al., 2013, p. 3 [44]), or backchannels (e.g., “is that right?”, “really?”). Questions intended for structuring, initiating or maintaining conversation (SIMCO; n = 74) include topic initiations (e.g., “So you go back over the summer, I assume?” [45]). Plans and actions questions (PlanAct; n = 53) suggest engaging in a particular action, proposing a future action or decision by the interlocutor, offers and invitations (e.g., “can I get you some coffee?” [46]). Following the same procedure as for question-response annotation, interrater reliability for the social action categories was calculated for 10% of the total number of question annotations (n = 686). Reliability between the two coders was calculated as 76% raw agreement and k = 0.70, again indicating substantial agreement. Due to having fewer than 100 occurrences each (i.e., fewer than the total number of dyads), we excluded SIMCOs and PlanAct from further analyses.

### 2.5. Kinematic Data Extraction

We extracted motion tracking data from three visual articulators: the torso (upper point between shoulders), head (top of head), and hands (center—both hands). These data were extracted from a 900 ms window, starting 300 ms prior to utterance onset, and ending 600 ms after utterance onset. We extracted the data in three 300 ms bins in order to preserve any variance in these features across the time window and include this information as a covariate in our models. Because the questions varied in length (see Appendix A Table A1), this time window includes sometimes just the beginning and sometimes up to the entire utterance. However, to keep the already rather complex analysis (including three different articulators, four kinematic measures and a set of six different social actions) as focused as possible, we here do not analyze movement by time bin but based on the 900 ms time window as a whole, the timeframe in which, from a cognitive processing perspective, social action recognition tends to take place [23,24]. More fine-grained and relative temporal analyses would go beyond the scope of the current study. However, by taking the utterances’ total duration into account in our models, this approach still allows us to determine whether bodily kinematics are associated with social action category within this 900 ms time window, and ensures that any association is unlikely to be explained by just interpersonal differences or differences in utterance duration between the categories.

Within the specified time window, we calculated four kinematic features. Magnitude captures the overall space in which the articulator moved during a time bin. In other words, it provides the maximum distance of the tracked point from its origin (i.e., its location at the beginning of the time bin). Lateral Position captures how much an articulator moves to the left or right during a time bin. Higher absolute values indicate further away from center along the *x*-axis. Anterior/Posterior Position captures how much a visual articulator moves towards or away from the interlocutor during a time bin. Higher values therefore indicate that the speaker (or more specifically, the articulator) is relatively closer, while lower values mean the speaker is relatively further away along the *z*-axis. Note that these values are with reference to the distance between participant and the Kinect device, but transformed in orientation, because the Kinect was physically at an angle from the participant the proximity value indicates the distance directly in front of or behind the participant. Peak Velocity captures the maximum velocity of the articulator within the time window. This value is taken as the derivative of absolute displacement, meaning that velocity is always positive. For the hands, these values are calculated as the maximum (in the case of magnitude, lateral position and peak velocity), or minimum (i.e., most proximal—in the case of anterior/posterior position) value achieved by the two hands. To avoid convergence issues due to differences in scale between model variables (see below for modeling procedure), we rescaled the proximity values based on standard deviation.

All data extraction and kinematic feature calculation was done using custom scripts running in Python (v3.7). Interfacing between Python and Elan annotations was done using the module pympi-ling [47].

### 2.6. Analyses

We used two sets of linear mixed models for our analyses of interest. The kinematic features described above were the dependent variables. All models included time bin as well as utterance duration as a covariate. Time bins were included in the model in order to account for fluctuations within the time window that would be lost when summing or averaging over the entire time window (e.g., peak velocity may be very high at one particular point but low across the rest of the window—in this case, only one time bin would show a high peak velocity, whereas that may not be representative of the entire window). Utterance duration was included in order to ensure that any inherent differences in utterance length between or within social action categories could not account for kinematic differences. We additionally tried to fit models with pair/participant as a nested random intercept, but otherwise used ‘file’ (i.e., a unique identifier for each participant) when convergence issues were encountered. Random slopes were modeled whenever this did not result in model convergence failures, singular fits, etc. This null model, containing only random terms and the covariates described above, was compared against a model of interest that also included social action category as a predictor variable, and when possible, a per-participant random slope. When not possible, this is noted in the results. Model comparison was done with a chi-squared test. In the case of significant model fit for the model of interest, we used post hoc contrasts to determine which social action categories differed from one another. *p*-values for these contrasts were adjusted using the Tukey method for comparing a family of six (i.e., the number of social action categories) conditions, across 15 comparisons. All analyses were performed in R [48], with the lme4 package used for the linear mixed models [49] (p. 4), emmeans used for post hoc comparisons [50], and sjPlot [51] and raincloud plots [52] used for visualization of the results.

## 3. Results

### 3.1. Torso Movements

In terms of torso movements, we found no evidence for anterior/posterior position being affected by social action category, χ^2^(5) = 11.007, *p* = 0.051. However, we did find evidence for a difference in torso movement size, χ^2^(5) = 31.76, *p* < 0.001. Specifically, we found that Information Requests (estimate = 0.12 ± 0.04, *z*-ratio = 3.292, *p* = 0.013), Understanding Checks (estimate = 0.16 ± 0.04 mm, *z*-ratio = 4.156, *p* < 0.001), Stance or Sentiment questions (estimate = 0.21 ± 0.04 mm, *z*-ratio = 5.243, *p* < 0.001), and Self-Directed questions (estimate = 0.14 ± 0.04 mm, *z*-ratio = 3.610, *p* = 0.004) all showed larger torso movements compared to Active Participation questions, while torso movements accompanying Stance or Sentiment questions were also larger than Information Requests (estimate = 0.09 ± 0.03 mm, *z*-ratio = 3.313, *p* = 0.012) as well as other initiated repairs (estimate = 0.13 ± 0.04, *z*-ratio = 3.165, *p* = 0.019). We additionally found evidence for lateral movement being affected by social action category, χ^2^(5) = 12.681, *p* = 0.027. However, no individual contrasts were significant. Finally, we also found peak velocity to be affected by social action category, χ^2^(5) = 40.054, *p* < 0.001. We specifically found that torso movements accompanying Information Requests had a higher peak velocity than those accompanying Other-Initiated Repairs (estimate = 1.373 ± 0.47 cm/s, *z*-ratio = 2.905, *p* = 0.043), those with Self-Directed Questions had higher peak velocity than those with Information Requests (estimate = 1.24 ± 0.34, *z*-ratio = 3.708, *p* = 0.003) as well as Understanding Checks (estimate = 1.19 ± 0.38, *z*-ratio = 3.167, *p* = 0.019), Other-Initiated Repairs (estimate = 2.61 ± 0.51, *z*-ratio = 5.14, *p* < 0.001) and Active Participation questions (estimate = 2.34 ± 0.47, *z*-ratio = 4.987, *p* < 0.001), while torso movements accompanying Stance or Sentiment questions had higher peak velocity than those accompanying Active Participation questions (estimate = 1.599 ± 0.50, z-ratio-3.183, *p* = 0.018). See Figure 2 for an overview of these findings. No model reported here contained random slopes.

### 3.2. Head Movements

In terms of head movements, we found that anterior/posterior position was significantly related to social action category, χ^2^(25) = 18.908, *p* = 0.002. Post-hoc comparisons showed that head movements accompanying Understanding Checks had closer proximity (i.e., larger A/P position values) compared to those accompanying Requests for Information (estimate = 0.13 ± 0.03, *z*-ratio = 4.047, *p* < 0.001). Note that effect sizes cannot be directly interpreted due to these values being scaled to facilitate modeling. We found no evidence for size of movement being related to social action category, χ^2^(5) = 8.654, *p* = 0.124. This magnitude model did not contain random slopes. We found no evidence for lateral position of the head being associated with social action category, χ^2^(5) = 4.769, *p* = 0.445. Finally, we found that peak velocity of the head was associated with social action category, χ^2^(5) = 11.607, *p* = 0.041. However, we found no significant contrasts between the pairs of categories. See Figure 3.

### 3.3. Manual Movements

In terms of hand movements, we found that anterior/posterior position was significantly related to social action category, χ^2^(5) = 27.895, *p* < 0.001. Specifically, we found that Active Participation questions were more proximal than other-initiated-repairs (estimate = 0.17 ± 0.05, *z*-ratio = 3.126, *p* = 0.022), Self-Directed questions (estimate = 0.18 ± 0.04, *z*-ratio = 4.084, *p* < 0.001) and Understanding Checks (estimate = 0.13 ± 0.04, *z*-ratio = 3.009, *p* = 0.032), while Stance or Sentiment questions were more proximal than Self-Directed questions (estimate = 0.15 ± 0.04, *z*-ratio = 3.871, *p* = 0.001). Note that effect sizes cannot be directly interpreted due to these values being scaled to facilitate modeling. We additionally found that the size of hand movements was significantly related to social action category, χ^2^(5) = 14.687, *p* = 0.012. Specifically, we found that hand movements with Understanding Checks were larger than with Active Participation questions (estimate = 0.98 ± 0.27 mm, *z*-ratio = 3.626, *p* = 0.004). We found no association between lateral position of the hands and social action category, χ^2^(5) = 10.336, *p* = 0.066. Finally, we found that peak velocity of the hands was associated with social action category, χ^2^(5) = 13.662, *p* = 0.017. Specifically, hand movements with Understanding Checks had higher peak velocity than those with Active Participation questions (estimate = 8.554 ± 2.80, *z*-ratio = 3.054, *p* = 0.027). See Figure 4 for an overview of these results.

None of these models contained random slopes.

## 4. Discussion

Our findings provide the first quantitative evidence for movement of the torso, head, and hands kinematically signaling specific social action categories during conversation. Specifically, we found that movement magnitude of the torso and hands differed between several categories, as did proximity of the hands to the addressee. Proximity of the head was also related to social action category, although no specific contrasts between categories survived statistical correction.

Our finding of torso, head and hand movements being related to social action categories suggests that, similar to concrete (e.g., grasping to drink or to pour) and more coarse-grained social intentions (e.g., the intention to communicate), what we intend to do with a question also influences the way that we move while producing the utterance. While this study was largely exploratory due to the novelty of our research question, we can see several patterns emerging from our results. For example, active participation questions (e.g., a backchannel “really?”) show much smaller movement magnitudes of the torso compared to other categories. While one may suggest that such effects may also be confounded by differences in duration (i.e., requests for information are likely to be, on average, longer than active participations), this was controlled for by including the total utterance duration in the null model. Therefore, in the case of significant models, social action category explained more variance in the kinematic data than just the differences in utterance duration. We would speculate that active participation, at least in the case of backchannel questions, may be accompanied by smaller magnitude movements and lower peak velocity for the very reason they occur in the backchannel and are therefore intended to attract less visual attention, which movement of prominent visual articulators such as the torso would otherwise add. It is also possible, however, that questions expressing active participation are associated with other visual signals not captured here, such as facial signals.

Stance or sentiment questions (e.g., “isn’t that quite far away?”—potentially indicating an affective response to the previous speaker’s plan or suggestion in the way they are embedded in the interaction), were found to be accompanied by the largest torso movements, with significantly greater magnitude than requests for information, active participations, and other-initiated repairs. While speculative, this may due to the affective nature of such utterances, given that affect can also influence bodily movement [53], which may involve moving the torso forwards or backwards to express such evaluative or affiliative stances, including distancing oneself from a statement or showing ones agreement with it.

The relatively small torso movement magnitudes associated with other-initiated-repair questions may relate to the so-called “freeze-look” that speakers sometimes use. The freeze-look refers to when a speaker briefly “freezes” their posture while fixing their gaze on their interlocutor, which is particularly salient when the person providing the freeze-look was expected to provide a response to an interlocutor’s question. This phenomenon, which has been described both for spoken [54,55] as well as signed languages [55,56], may thus provide a visual signal through a lack of movement.

Results from our analysis of head motion suggest that proximity to the addressee is also influenced by the category of social action. However, we only found one significant contrast between categories. Specifically, understanding checks showed closer proximity (i.e., larger anterior/posterior position values) than requests for information. This may relate to a general signal of thrusting the head forward when expressing uncertainty [43,57,58]. The fact that we did not find a similarly strong effect of other-initiated-repairs despite them also expressing a lack of understanding and having been associated with forward thrusts in past research [57,58] is intriguing and may hint at an interesting interplay between articulators; other-initiated repairs are often signaled facially, for example [59,60]. However, the fact that other social action categories do not differ in head movement proximity may also suggest that such a signal may be commonly used across most question categories. Additionally, our finding of relatively few significant contrasts between social actions when looking at head movement kinematics, despite the extensive literature showing that head movements are involved in dyadic coordination [61,62,63], may be due to a difference in roles. More specifically, it may be that head movement kinematics are so tightly linked to torso movements, that we do not see distinct effects in the head, at least at this level of analysis. Future studies should investigate whether other factors within the interaction may play a role which may lead to more fine-grained analyses that shed further light on these issues.

Results from hand motion kinematics show several interesting, somewhat unexpected effects. We see that, while repair initiations have been previously linked to movements towards the addressee (e.g., leaning forward, thrusting the head forward [43,58,64]), other-initiated repairs in our study showed relatively lower movement magnitudes of the torso in comparison to stance or sentiment questions, and the hands were also further from the addressee during other-initiated repairs than during information requests, stance or sentiment questions, and active participation questions. One possible interpretation of this finding is that individuals in our study simply did not move their torso or hands closer to their addressee during other-initiated repair questions. However, previous studies have qualitatively described these movement effects in individual examples of other-initiated repairs, rather than contrasting them with other social actions. Therefore, a more likely explanation for our finding is that the other categories we have compared them to here simply show stronger proximity effects than other-initiated repairs. In this interpretation, other-initiated repairs may indeed be associated with movements that bring visual articulators of the repair initiator closer to the interlocutor, as has been described in the literature before [43,58,64]. However, this form of increase in proximity, by moving anteriorly towards the interlocutor, may be even more strongly employed for other social actions, such as requests for information, stance or sentiment questions, and active participation questions. Manual gestures may also be produced with closer proximity to the interlocutor (i.e., reaching farther into interpersonal space) when discussing shared goals or plans [65]. This is particularly interesting given that active participation questions conversely showed quite low movement magnitude and peak velocity in the torso. This finding is further discussed below. Importantly, proximity of articulators to the interlocutor is likely most salient as a signal of social action when it is a (relatively sudden) deviation from a previously less-proximal position. While our current results provide evidence for systematic differences between social action categories, an interesting avenue for future research would be to investigate how proximity dynamically changes, especially in the case of other-initiated repair (such as when comparing torso or hand position before and after onset of the utterance).

We additionally found that active participation questions weref accompanied by lower magnitude hand movements, at least compared to questions that check for understanding. This is particularly interesting since the hands showed closer proximity in active participation questions compared to understanding checks. This suggests that while the hands (start to) produce larger movements during understanding checks, this movement does not seem to reduce the distance between speaker and addressee. Although this is only one contrast, this finding also more generally fits with the finding from the torso movements, which similarly showed active participation questions having lower magnitude than several other categories, and the speculative interpretation relating to their backchannel status we have offered in connection with the torso movements above may also hold for manual movements. The interpretation of active participation questions as being less ‘urgent’ in the conversation, such as the case of backchannels, is also supported by the low movement magnitude and peak velocity of the torso during active participation questions. This could be indicative of speakers positioning themselves closer to the interlocutor during, or immediately before these questions, but not using large, salient movements to do so.

Our results are therefore in line with the idea that our intentions, even at the level of fine-grained social actions occurring in conversation, are (partially) revealed in the way that we move. A fruitful direction for future research would be to determine to what extent these kinematic differences indeed serve as information for the addressee in such a way that they may facilitate quick social action recognition. It is important to note that, just as in previous studies of intentions and kinematics, we do not assume that the kinematic parameters themselves (e.g., size, proximity) are sufficient to recognize the intention. Instead, the specific kinematics likely guide our interpretation. For example, in Trujillo and colleagues’ [14,66] work showing that communicative intentions could be recognized from gesture kinematics, observers accomplished this recognition based on how the kinematics differed within a specified form, and within the context of two potential intentions. In other words, the form (i.e., the action itself) must be recognized first, and the specific kinematic profile of the movements guided interpretation of the associated intention within the context of the experiment [14,66]. Similarly, in the case of social actions, we take these kinematic features to be systematic changes of whatever the person is doing while producing the utterance. If these kinematic changes indeed facilitate social action recognition for observers, this is likely indeed a facilitation effect, rather than completely and independently revealing the social action. In other words, social action recognition will likely still require holistic information about the unfolding movement, as well as the interactional context, speech, prosody, or all of the above. The recognition of the social action, thus already somewhat constrained or informed by these other factors, may then be further facilitated by integrating this kinematic information, and/or vice versa.

An interesting question for future research is to bring these findings together with the field of interpersonal synchrony, which has shown that dyads synchronize their bodily movements throughout an interaction, even at a millisecond time-scale. Our findings suggest that any effects of social actions on movement kinematics must occur on top of such synchrony patterns, or may otherwise temporarily disrupt patterns of synchrony, because social actions are often complementary in nature (giving-receiving, requesting-offering, inviting-declining, questioning-responding etc.), and this would likely be apparent in their corresponding movements (as corroborated by the present findings). How these two kinematic dynamics (i.e., ongoing synchrony and the effect of social actions) interact and influence one another, is an open question that would be informative for better understanding the fine-grained dynamics of interpersonal movement coordination, and how intention signaling fits into, and is shaped by, the larger dynamic of social interaction (as well as how variation in interpersonal synchrony may affect this, e.g., [67,68]). As part of such a future endeavor, looking in detail at the reciprocal relationship between interlocutors, and how one interlocutor’s social action kinematics influences those of the other’s social action kinematics (e.g., as has been shown for kinematic variations of reciprocal actions [69]) would be a highly interesting extension of the current work. Finally, future research should also investigate whether these patterns of social action kinematics generalize across neurodiverse populations, such as autistic individuals.

### 4.1. Limitations

The interpretations of our findings discussed above come with several limitations. First, these findings come from a corpus of conversation where many layers of behaviour are intertwined and influence one another. Therefore, we cannot draw any causal conclusions about these relationships. However, this paradigm allows a more ecological starting point for investigating this interesting, and thus far under-researched question. A second potential limitation is that our analyses do not take into account what the bodily action is that our kinematics are measuring. In other words, we cannot draw inferences about particular gesture types, specific forms of torso leans, particular head movements, etc. However, the kinematic information within this very early window (i.e., between 300 ms prior to utterance onset and 600 ms after onset) still provides interesting insights into the minimal information that is available to an interlocutor who is attempting fast recognition of the social action being performed by the current speaker. Further research should investigate whether these different social action categories are associated with the initiation of particular bodily actions, or whether these kinematic differences may relate to more subtle movements that systematically differ between categories. Future studies are also needed to look into more detail at how early exactly the kinematic signals occur with respect to the relative length of the utterances and the time by which a response would be required to be able to link it more informatively to early social action recognition in conversation [24]. The current study also does not account for different levels of familiarity between interlocutors, which may affect communicative behaviors [70]. A final limitation is that the scarcity of past research on the topic of movement kinematics in connection with social actions makes it difficult to draw strong causal conclusions regarding the differences between categories. This means that, for example, it is currently difficult to determine what a meaningfully large effect size would be that affects how the listener perceives the ongoing utterance. Future studies utilizing social action attribution paradigms will be needed to shed more light on this issue. While this means that our study is somewhat exploratory in nature, we believe that the main effect of several different articulator kinematics differing between categories is a very useful starting point for future research to explore these differences further. Both more fine-grained quantitative corpus analyses taking into account movement types and interactional context, as well as experiments testing for causal relations would be a fruitful continuation from the present study.

### 4.2. Conclusions

In sum, our study is the first to investigate how, during naturalistic conversation, movement of several prominent visual articulators (i.e., the torso, head, and hands) is shaped by the social action being performed by an utterance. These findings suggest that the kinematic expression of intentions carries over into the fine-grained social actions that our utterances perform in conversation, and thus provide a foundation for future experimental research and the development of situated models of language processing as well as social robots.

## Figures and Tables

**Figure 1 brainsci-11-00996-f001:**
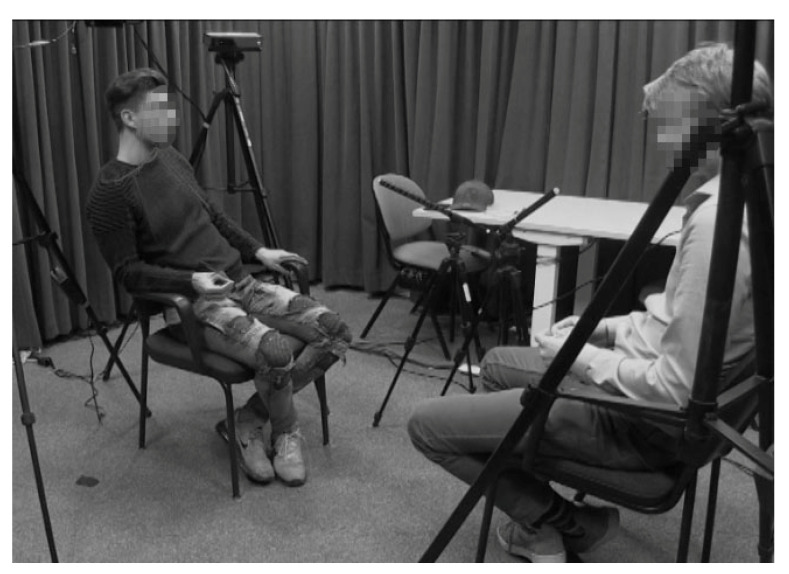
View of recording set-up as seen from participant A’s (left) camera. Visible in this still frame is also one of the Microsoft Kinects, behind participant A’s left shoulder, as well as the tripods for two of the other Kinects, and the two microphones (center).

**Figure 2 brainsci-11-00996-f002:**
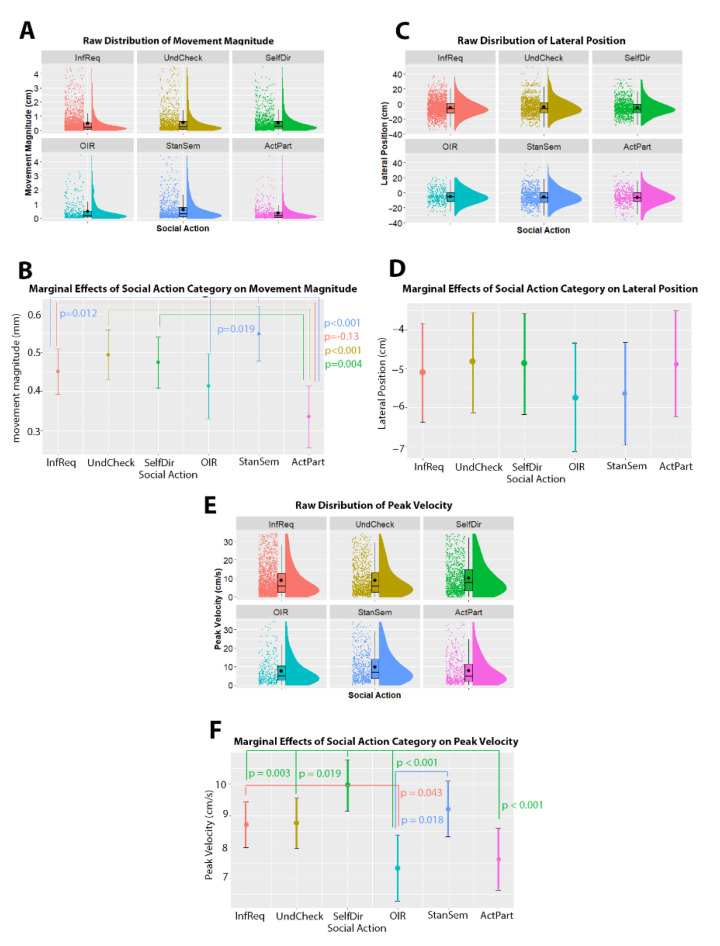
Relationship between torso movement and social action category. Panels (**A**,**C**,**E**) show raincloud plots of the raw distribution (*y*-axis) across each of the social action categories (*x*-axis). The raincloud plots show the overall density of the distribution (shaded curve), the boxplots show the median and interquartile range, while the black dots show the individual data points. Panels (**B**,**D**,**F**) depicts the marginal effects of the mixed model (i.e., the predicted values when holding other model terms, such as utterance duration and per-participant intercepts, constant). In panels (**B**,**D**,**F**) social action categories are given on the *x*-axis, and kinematic values are given on the *y*-axis. Significant differences between pairs of categories are indicated with colored lines between the categories. *p*-values are given for each of these comparisons, with the text color corresponding to the larger value in the comparison. Abbreviations: InfReq = Information Request; UndCheck = Understanding Check; SelfDir = Self-Directed Question; OIR = Other-Initiated-Repair; StanSem = Stance or Sentiment; ActPart = Active Participation.

**Figure 3 brainsci-11-00996-f003:**
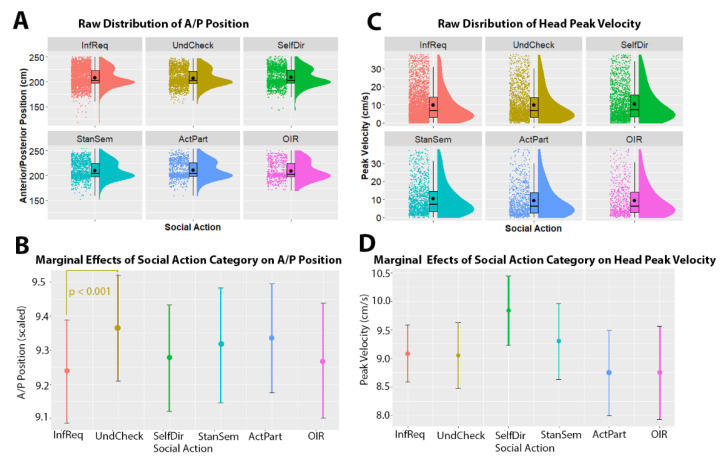
Relationship between head movement and social action category. Panels (**A**,**C**) show raincloud plots of the raw distribution of kinematic values (*y*-axis) across each of the social action categories (*x*-axis). The raincloud plots show the overall density of the distribution (shaded curve), the boxplots show the median and interquartile range, while the black dots show the individual data points. Panels (**B**,**D**) depict the marginal effects of the mixed model (i.e., the predicted values when holding other model terms, such as utterance duration and per-participant intercepts, constant). In panels (**B**,**D**), social action categories are given on the *x*-axis, and kinematic values are given on the *y*-axis (note that anterior/posterior position is scaled to range = 0.1–12.7) is given on the *y*-axis. Note that in this model, there were no significant contrasts between social action categories. Abbreviations: InfReq = Information Request; UndCheck = Understanding Check; SelfDir = Self-Directed Question; OIR = Other-Initiated-Repair; StanSem = Stance or Sentiment; ActPart = Active Participation.

**Figure 4 brainsci-11-00996-f004:**
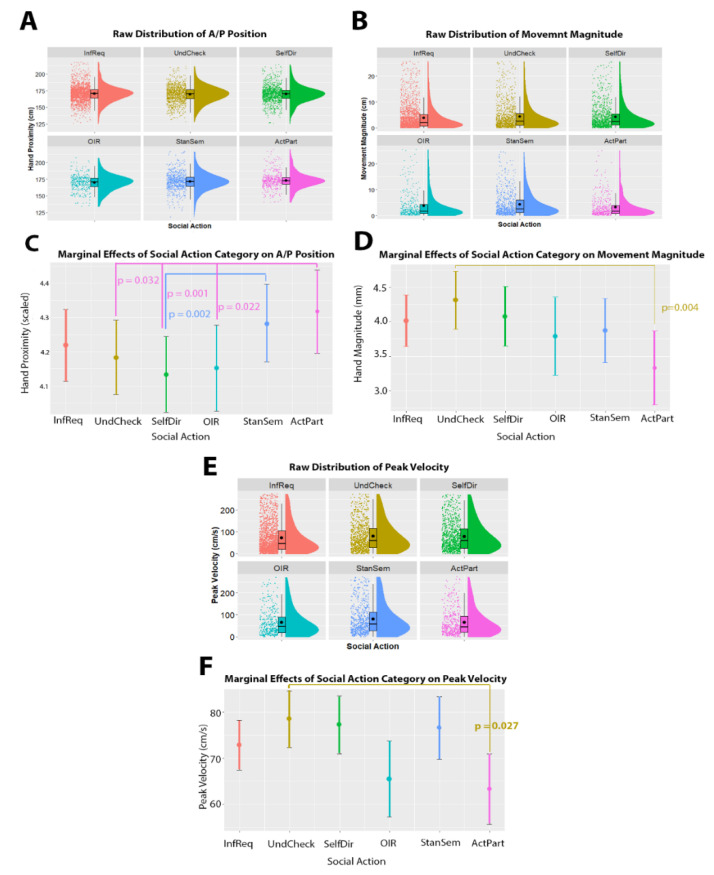
Relationship between hand movement and social action category. Panels (**A**,**C**,**E**) show raincloud plots of the raw distribution (*y*-axis) across each of the social action categories (*x*-axis). The raincloud plots show the overall density of the distribution (shaded curve), the boxplots show the median and interquartile range, while the black dots show the individual data points. Panels (**B**,**D**,**F**) depicts the marginal effects of the mixed model (i.e., the predicted values when holding other model terms, such as utterance duration and per-participant intercepts, constant). In panels (**B**,**D**,**F**) social action categories are given on the *x*-axis, and kinematic values are given on the *y*-axis Note that anterior/posterior position is scaled to range 0.1–15.3. Significant differences between pairs of categories are indicated with colored lines between the categories. *P*-values are given for each of these comparisons, with the text color corresponding to the larger value in the comparison. Abbreviations: *InfReq* = Information Request; *UndCheck* = Understanding Check; *SelfDir* = Self-Directed Question; *OIR* = Other-Initiated-Repair; *StanSem* = Stance or Sentiment; *ActPart* = Active Participation’.

## Data Availability

Raw data for this study are not available due to privacy of participants. Analysis and processing scripts are available on OSF at: https://osf.io/w9ms4/ (accessed on 27 July 2021).

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
