# Peer review of "The Kinematics of Social Action: Visual Signals Provide Cues for What Interlocutors Do in Conversation"

_brainsci, 2021, doi:10.3390/brainsci11080996_

Round 1

Reviewer 1 Report

The authors Trujillo and Holler present a study examining the effects of different social action types on the movement (in terms of proximity, and movement during) of the head and hands during natural conversations between dyads of observers. This manuscript reports on a study using of a corpus of video, audio, and motion-tracked conversations. The question being asked in the present manuscript is an extremely interesting one, relating to whether motor behaviours might be used as cues to a conversational partner from a very early time while asking questions (allowing the observer to begin to prepare their response). I thought that this is an extremely worthwhile question to ask, however, I had quite several concerns regarding the method and analysis of this research that I believe need to be clarified to be confident to recommend publication. It may be that the researchers have excellent answers to my comments, and I sincerely hope that they do. I think that this has the potential to be an excellent paper, so would be delighted to be put ‘back in my box’! With that in mind, I hope that the authors read my review with the constructive tone in which it is intended.

MAJOR COMMENTS

Analysis of time-course information

On page 6, lines 272-275 you mention that data was extracted into three time-bins (of 300ms). I could not tell from your analyses how the binning of data was treated. This is a shame, especially as you talk about the time-course of those kinematics being really important. It seems more (from my reading of your analyses) that you have just analysed datapoint for each participant, for each utterance. Can you please unpack why you looked at time bins, and how this was treated in analysis?

The role of the dyads

Did you consider including the dyad as a random-effect in your lme models? I would imagine that different dyads might have different reliance on physical movement for the conversation. Further, you mention that the dyads were ‘acquaintances’ (page 4, line 154). I feel like more information would be required here. Would you expect friends who have known each other for a long time to behave the same as friends who have only recently met? (e.g., see Feyereisen, P. (1994). The behavioural cues of familiarity during social interactions among human adults: A review of the literature and some observations in normal and demented elderly subjects. Behavioural processes33(1-2), 189-211.). Similarly, did you consider the gender makeup of the dyads, particularly as gender may affect some elements of gesturing (Skomroch, H., Petermann, K., Helmich, I., Dvoretska, D., Rein, R., Kim, Z. H., & Lausberg, H. (2013). Gender differences in hand movement behavior. In Poster presented at the Tilburg Gesture Research Meeting (TiGeR), Tilburg, The Netherlands. Retrieved from tiger. uvt. nl/pdf/papers/skomroch. pdf.).  

The types of conversation

You discuss the different types of conversations that people had (page 4, lines 158-165). I was not clear why you asked them to complete different conversations, or whether you thought that might affect the results for this study. I ask this particularly as the topic of conversation might influence gesturing (and therefore hand movement) (Sundstrom, E. (1978). A test of equilibrium theory: Effects of topic intimacy and proximity on verbal and nonverbal behavior in pairs of friends and strangers. Environmental psychology and nonverbal behavior3(1), 3-16.).

The validity of the Kinect recording

One major concern I had with this work was that you do not really comment on the spatial and temporal validity of the Kinect and Brekel packages. I appreciate that you synched the signals with a clap (line 189), but I would like more information to confirm that there could not have been a temporal lag, and that the spatial resolution of the system was enough to discern the differences that you have observed which seem to be quite subtle in magnitude, and across quite short timescales.

Duration of social actions

You note that there were 6778 questions extracted in total (line 222), but then cut that to 2078 social action questions (line 226). I was not clear on the duration of these social action questions (despite it being outlined for the question pool in whole in lines 222-223), but I thought that information would be very important. Similarly, I would like to see the distributions of the utterance lengths for the different categories to check that there was not a systematic difference. I appreciate that you included this in the model, but I would like to see that this was not a systematic difference.

The influence of outliers

One thing that is quite evident in your graphs, but I found a little confusing was the outliers (e.g., at zero in Figure 3A on page 9, and on Figure 3A (a typo?) on page 10 [again at 0, perhaps with a flipped x-axis?]). Outliers themselves are perhaps not surprising, but that there is nothing at all between 0 and 100 made me wonder whether these data points are real? For example – the data clusters quite nicely at around 250mm in Figure 3A (page 9), with perhaps some negative skew or bimodality in InfReq or UndCheck. These outliers are highly extreme. Can you please explain what these are? If they are valid datapoints, can you also examine whether your results are consistent with these values removed? If so – this can be as simple as a statement to say you have checked, but would give me more confidence in your results.

Further to this – I would imagine that you need to check the residuals of your lme models (e.g., like this: https://stats.stackexchange.com/questions/77891/checking-assumptions-lmer-lme-mixed-models-in-r) to ensure that you meet the assumptions for modelling like this. Those outliers might be a problem.

p-values in LME models

I had a few questions regarding your use of p-values. This is more of a general comment on overreliance on p-values, but one that I think relates here. The binary significant/non-significant cut-off of 0.05 is something that is so arbitrary, I think it is problematic that you ‘accept’ null hypotheses for p-values that do not quite cross this threshold (e.g., with proximity having a p = .051 on line 316) but reject them for ones that are just under (e.g., on line 344 where p = .04) or way under (e.g., p < .001). I think effect sizes would be way more valuable to the reader here. There are some nice guides how to get that from lme models here https://easystats.github.io/effectsize/articles/from_test_statistics.html, which use the lmerTest package, and runs anova() functions on the models – so slightly different from the model comparison approach you have used here, but also provides 95% CI for the effects.

MINOR COMMENTS

Line 15 ‘in conversation is’ should read ‘in conversation are

In line 181 you refer to an Appendix – is this Figure 1? I did not get an appendix with the manuscript download.

Line 198 ‘Participantion was…’ should read ‘Participation was…

In line 308 you refer to Tukey corrections for a ‘family of six’ estimates. I assume here you meant correcting for the 15 comparisons between the 6 conditions (not a correction for 6 comparisons). I think given your use of r for this that you have selected Tukey corrections, rather than manually making any adjustments, but that might just be tidied up in the language you have used.

For your graphs (and I was delighted to see some wonderful use of ggplot!), it might be better to use facet_grid() to separate out the Social Actions in the upper-panels. I suggest this as the colours would be lost in b&w printing (how I know many academics print things), but also that the order of your rainclouds (top to bottom) was opposite to your legend. Faceting (even with colour) would ensure that your graphs would be equally legible in bw or colour. You could also flip the coordinates of the upper-panel to make the social actions vertically-match the lower-panels.

Line 360-361 seemed to have been rendered in bold.

Reviewer 2 Report

This work addresses a very interesting and important topic. That is, how kinematics are shaped by the fine-grained intentions underlying the social actions of conversation. Although his aim is stated at the end of the introduction (lines: 149-150), from the abstract and the earlier parts of the Introduction it seems like the focus is on a specific movements. The authors dedicated an in depth discussion on the link between specific movements and used in particular social action. Given the fact that the current study focuses on kinematic feature and not specific movements, I think that this parts should not be the focus of the introduction.

My major concern is the small sample size, which can be easily addressed by providing a power calculation.

Notably, one strength of the current study is its naturalistic settings. While the authors measures the  kinematics of the movement of each participant, they ignore the reciprocal nature of naturalistic social interactions. For example, the kinematic features of one subjects are affected by the other. Moreover kinesthetic cues may fosters interpersonal synchrony, which is a core mechanism of social interaction. Tracking the movement of two participants simultaneously allows measuring behavioral synchrony in an objective automated and ecological manner.

Recent evidence shows that reduced is associated with deficits in social cognition, e.g., in in autism spectrum disorder (ASD see: https://www.nature.com/articles/s41598-020-74951-8  ) and with attention deficit hyperactivity disorder (ADHD see: https://www.sciencedirect.com/science/article/pii/S0001691820305345#bb0170 ).

These recent findings should be discussed.

Importantly, the exclusion criteria does not include ADHD. Is it possible that individuals with ADHD were included ?

Moreover kinematic features were recently found to be associated with autistic traits see:

Is it possible that individuals with high autistic traits  were included ?

The kincet systems have been previously used to detect human gestures in a naturalistic environment (Martin et al. 2012; Sung et al. 2011). Surprisingly, these previous studies have not been reviewed.

Importantly, the kincet was critiqued for its ability to capture small movements of the extremities (such as hand and foot). Can you please address this criticism and how you overcome this?

Is there a specific reason why the authors have chosen these specific kinematic features, and not others? I would expect the authors to choose kinematic features that were previously found to be related to related to empathy and attunement.

If I understood correctly, Proximity may be related to interpersonal spaceis that correct? If think that the authors should discuss this possibility.

Did you find different results for each of the three themes (privacy, social media, or language in teaching)?

Head motions were used to analyze non-verbal dyad interactions in conversations (for review Delaherche et al., 2012) and in dyadic dance (Boker et al., 2002). How do you explain why the head was not more informative than other parts?

Round 2

Reviewer 1 Report

I would like to thank the authors for their clear responses to my comments. I am satisfied that they have addressed all of my concerns and would recommend that this paper be published, and believe that it will make a valuable addition to the literature.

Reviewer 2 Report

The manuscript has been greatly improved, the authors have addressed all the comments and is now ready for the publication.